# Deep Surrogate Assisted Generation of Environments

**Varun Bhatt**[*]
University of Southern California
Los Angeles, CA
vsbhatt@usc.edu

**Bryon Tjanaka**[*]
University of Southern California
Los Angeles, CA
tjanaka@usc.edu

**Matthew C. Fontaine**[*]
University of Southern California
Los Angeles, CA
mfontain@usc.edu

**Stefanos Nikolaidis**
University of Southern California
Los Angeles, CA
nikolaid@usc.edu

## Abstract

Recent progress in reinforcement learning (RL) has started producing generally capable agents that can solve a distribution of complex environments. These agents are typically tested on fixed, human-authored environments. On the other hand, quality diversity (QD) optimization has been proven to be an effective component of environment generation algorithms, which can generate collections of high-quality environments that are diverse in the resulting agent behaviors. However, these algorithms require potentially expensive simulations of agents on newly generated environments. We propose Deep Surrogate Assisted Generation of Environments (DSAGE), a sample-efficient QD environment generation algorithm that maintains a deep surrogate model for predicting agent behaviors in new environments. Results in two benchmark domains show that DSAGE significantly outperforms existing QD environment generation algorithms in discovering collections of environments that elicit diverse behaviors of a state-of-the-art RL agent and a planning agent. Our source code and videos are available at https://dsagepaper.github.io/

## 1 Introduction

We present an efficient method of automatically generating a collection of environments that elicit diverse agent behaviors. As a motivating example, consider deploying a robot agent at scale in a variety of home environments. The robot should generalize by performing robustly not only in test homes, but in any end user's home. To validate agent generalization, the test environments should have good coverage for the robot agent. However, obtaining such coverage may be difficult, as the generated environments would depend on the application domain, e.g. kitchen or living room, and on the specific agent we want to test, since different agents exhibit different behaviors.

To enable generalization of autonomous agents to new environments with differing levels of complexity, previous work on open-ended learning [1, 2] has integrated the environment generation and the agent training processes. The interplay between the two processes acts as a natural curriculum for the agents to learn robust skills that generalize to new, unseen environments [3–5]. The performance of these agents has been evaluated either in environments from the training distribution [1, 2, 5] or in suites of manually authored environments [3, 6, 4].

As a step towards testing generalizable agents, there has been increasing interest in competitions [7, 8] that require agents to generalize to new game layouts. Despite the recent progress of deep learning

---

[*]Equal contribution

36th Conference on Neural Information Processing Systems (NeurIPS 2022).

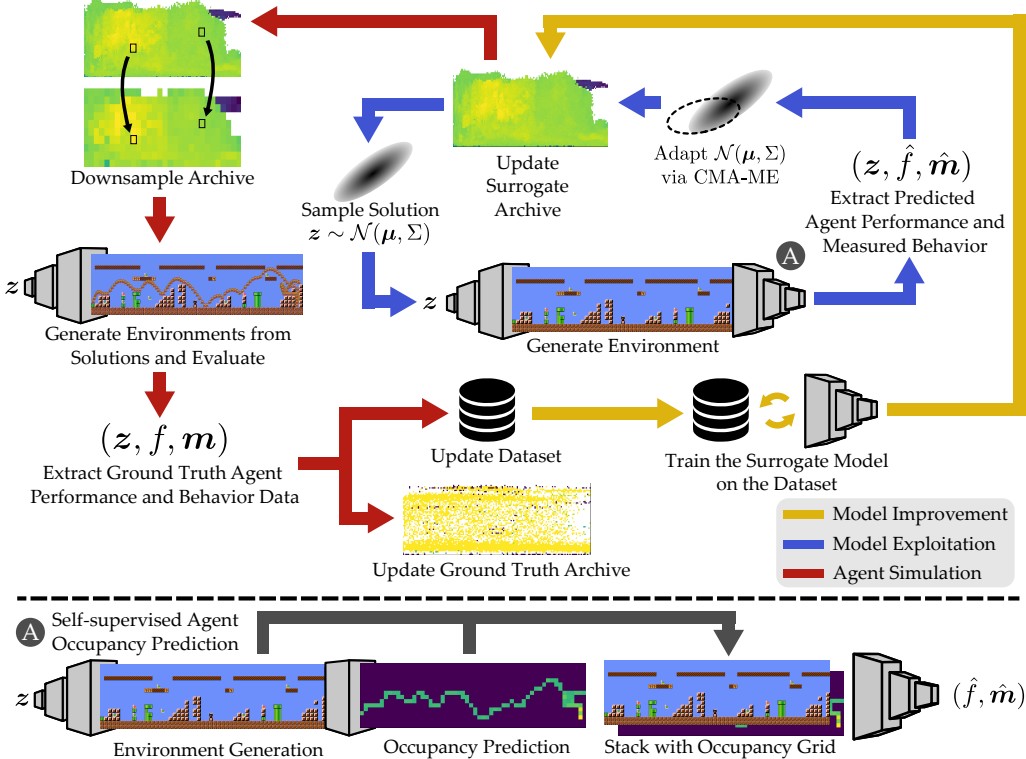

Figure 1: An overview of the Deep Surrogate Assisted Generation of Environments (DSAGE) algorithm. The algorithm begins by generating and evaluating random environments to initialize the dataset and the surrogate model (not shown in the figure). An archive of solutions is generated by exploiting a deep surrogate model (**blue arrows**) with a QD optimizer, e.g., CMA-ME [14]. A subset of solutions from this archive are chosen by downsampling and evaluated by generating the corresponding environment and simulating an agent (**red arrows**). The surrogate model is then trained on the data from the simulations (**yellow arrows**). While the images show Mario levels, the algorithm structure is similar for mazes.

agents in fixed game domains, e.g. in Chess [9], Go [10], Starcraft [11], and Poker [12, 13], it has been rule-based agents that have succeeded in these competitions [8]. Such competitions also rely on manually authored game levels as a test set, handcrafted by a human designer.

While manually authored environments are important for standardized testing, creating these environments can be tedious and time-consuming. Additionally, manually authored test suites are often insufficient for eliciting the diverse range of possible agent behaviors. Instead, we would like an interactive test set that proposes an environment, observes the agent's performance and behavior, and then proposes new environments that diversify the agent behaviors, based on what the system has learned from previous execution traces of the agent.

To address collecting environments with diverse agent behaviors, prior work frames the problem as a quality diversity (QD) problem [15–17]. A QD problem consists of an objective function, e.g. whether the agent can solve the environment, and measure functions, e.g. how long the agent takes to complete their task. The measure functions quantify the behavior we would like to vary in the agent, allowing practitioners to specify the case coverage they would like to see in the domain they are testing. While QD algorithms can generate diverse collections of environments, they require a large number of environment evaluations to produce the collection, and each of these evaluations requires multiple time-consuming simulated executions of potentially stochastic agent policies.

We study how *deep surrogate models that predict agent performance can accelerate the generation of environments that are diverse in agent behaviors*. We draw upon insights from model-based quality diversity algorithms that have been previously shown to improve sample efficiency in design optimization [18] and Hearthstone deckbuilding [19]. Environments present a much more complex

prediction task because the evaluation of environments involves simulating stochastic agent policies, and small changes in the environment may result in large changes in the emergent agent behaviors [20].

We make the following contributions: (1) We propose the use of deep surrogate models to predict agent performance in new environments. Our algorithm, Deep Surrogate Assisted Generation of Environments (DSAGE) (Fig. 1), integrates deep surrogate models into quality diversity optimization to efficiently generate diverse environments. (2) We show in two benchmark domains from previous work, a Maze domain [3, 4] with a trained ACCEL agent [4] and a Mario domain [21, 16] with an A* agent [22], that DSAGE outperforms state-of-the-art QD algorithms in discovering diverse agent behaviors. (3) We show with ablation studies that training the surrogate model with ancillary agent behavior data and downsampling a subset of solutions from the surrogate archive results in substantial improvements in performance, compared to the surrogate models of previous work [19].

## 2 Problem Definition

**Quality diversity (QD) optimization.** We adopt the QD problem definition from previous work [23]. A QD optimization problem specifies an objective function $f : \mathbb{R}^n \to \mathbb{R}$ and a joint measure function $\boldsymbol{m} : \mathbb{R}^n \to \mathbb{R}^m$. For each element $s \in S$, where $S \subseteq \mathbb{R}^m$ is the range of the measure function, the QD goal is to find a solution $\boldsymbol{\theta} \in \mathbb{R}^n$ such that $\boldsymbol{m}(\boldsymbol{\theta}) = s$ and $f(\boldsymbol{\theta})$ is maximized.

Since the range of the measure function can be continuous, we restrict ourselves to algorithms from the MAP-Elites family [24, 25] that discretize this space into a finite number of $M$ cells. A solution $\boldsymbol{\theta}$ is mapped to a cell based on its measure $\boldsymbol{m}(\boldsymbol{\theta})$. The solutions that occupy cells form an *archive* of solutions. Our goal is to find solutions $\boldsymbol{\theta}_i, i \in \{1, ..., M\}$ that maximize the objective $f$ for all cells in the measure space.

$$\max_{\boldsymbol{\theta}_i} \sum_{i=1}^{M} f(\boldsymbol{\theta}_i) \tag{1}$$

The computed sum in Eq. 1 is defined as the QD-score [26], where empty cells have an objective value of 0. A second metric of the performance of a QD algorithm is coverage of the measure space, defined as the proportion of cells that are filled in by solutions: $\frac{1}{M} \sum_{i=1}^{M} \mathbf{1}_{\boldsymbol{\theta}_i}$.

**QD for environment generation.** We assume a single agent acting in an environment parameterized by $\boldsymbol{\theta} \in \mathbb{R}^n$. The environment parameters can be locations of different objects or latent variables that are passed as inputs to a generative model [27].[2] A QD algorithm generates new solutions $\boldsymbol{\theta}$ and evaluates them by simulating the agent on the environment parameterized by $\boldsymbol{\theta}$. The evaluation returns an objective value $f$ and measure values $\boldsymbol{m}$. The QD algorithm attempts to generate environments that maximize $f$ but are diverse with respect to the measures $\boldsymbol{m}$.

## 3 Background and Related Work

**Quality diversity (QD) optimization.** QD optimization originated in the genetic algorithm community with diversity optimization [28], the predecessor to QD. Later work introduced objectives to diversity optimization and resulted in the first QD algorithms: Novelty Search with Local Competition [29] and MAP-Elites [25, 24]. The QD community has grown beyond its genetic algorithm roots, with algorithms being proposed based on gradient ascent [23], Bayesian optimization [30], differential evolution [31], and evolution strategies [14, 32, 33]. QD algorithms have applications in damage recovery in robotics [24], reinforcement learning [34–36], and generative design [18, 37].

Among the QD algorithms, those of particular interest to us are the model-based ones. Current model-based [38, 39] QD algorithms either (1) learn a surrogate model of the objective and measure functions [18, 40, 41], e.g. a Gaussian process or neural network, (2) learn a generative model of the representation parameters [42, 43], or (3) draw inspiration from model-based RL [44, 45]. In particular, Deep Surrogate Assisted MAP-Elites (DSA-ME) [19] trains a deep surrogate model on a diverse dataset of solutions generated by MAP-Elites and then leverages the model to guide MAP-Elites. However, DSA-ME has only been applied to Hearthstone deck building, a simpler prediction problem than predicting agent behavior in generated environments. Additionally, DSA-ME is specific to MAP-Elites only and cannot run other QD algorithms to exploit the surrogate model.

---

[2]For consistency with the generative model literature, we use $\mathbf{z}$ instead of $\boldsymbol{\theta}$ when denoting latent vectors

Furthermore, DSA-ME is restricted to direct search and cannot integrate generative models to generate environments that match a provided dataset.

**Automatic environment generation.** Automatic environment generation algorithms have been proposed in a variety of fields. Methods between multiple communities often share generation techniques, but differ in how each community applies the generation algorithms.

For example, in the procedural content generation (PCG) field [46], an environment generator produces video game levels that result in player enjoyment. Since diversity of player experience and game mechanics is valued in games, many level generation systems incorporate QD optimization [47, 16, 48–52]. The procedural content generation via machine learning (PCGML) [53, 54] subfield studies environment generators that incorporate machine learning techniques such as Markov Chains [55], probabilistic graphical models [56], LSTMs [57], generative models [58–61], and reinforcement learning [62, 63]. Prior work [64] has leveraged surrogate models trained on offline data to accelerate search-based PCG [65].

Environment generation methods have also been proposed by the scenario generation community in robotics. Early work explored automated methods for generating road layouts, vehicle arrangements, and vehicle behaviors for testing autonomous vehicles [66–72]. Outside of autonomous vehicles, prior work [73] evaluates robot motion planning algorithms by generating environments that target specific motion planning behaviors. In human-robot interaction, QD algorithms have been applied as environment generators to find failures in shared autonomy systems [17] and human-aware planners tested in the collaborative Overcooked domain [15]

Environment generation can also help improve the generality of RL agents. Prior work proposes directly applying PCG level generation algorithms to improve the robustness of RL [74, 75] or to benchmark RL agents [76]. Paired Open-ended Trailblazer (POET) [1, 2] coevolves a population of both agents and environments to discover specialized agents that solve complex tasks. POET inspired a variety of open-ended coevolution algorithms [77–79, 5]. Later work proposes the PAIRED [3], PLR [80, 6], and ACCEL [4] algorithms that train a single generally capable agent by maximizing the regret between a pair of agents. These methods generate environments in parallel with an agent to create an automatic training curriculum. However, the authors validate these methods on human-designed environments [81]. Our work proposes a method that automatically generates valid environments that reveal diverse behaviors of these more general RL agents.

## 4 Deep Surrogate Assisted Generation of Environments (DSAGE)

**Algorithm.** We propose the Deep Surrogate Assisted Generation of Environments (DSAGE) algorithm for discovering environments that elicit diverse agent behaviors. Akin to the MAP-Elites family of QD algorithms, DSAGE maintains a *ground-truth archive* where solutions are stored based on their ground-truth evaluations. Simultaneously, DSAGE also trains and exploits a deep surrogate model for predicting the behavior of a fixed agent in new environments. The QD optimization occurs in three phases that take place in an outer loop: model exploitation, agent simulation, and model improvement (Fig. 1). Algorithm 1 provides the pseudocode for the DSAGE algorithm.

The model exploitation phase (lines 11–20) is an inner loop that leverages existing QD optimization algorithms and the predictions of the deep surrogate model to build an archive – referred to as the *surrogate* archive – of solutions. The first step of this phase is to query a list of $B$ candidate solutions through the QD algorithm's *ask* method. These solutions are environment parameters, e.g., latent vectors of a GAN, which are passed through the environment generator, e.g., a GAN, to create an environment (line 15). Next, we make predictions with the surrogate model. The surrogate model first predicts data representing the agent's behavior, e.g., the probability of occupying each discretized tile in the environment (line 16), referred to as "ancillary agent behavior data" ($y$). The predicted ancillary agent behavior data ($\hat{y}$) then guides the surrogate model's downstream prediction of the objective ($\hat{f}$) and the measure values ($\hat{m}$) (line 17). Finally, the QD algorithm's *tell* method adds the solution to the surrogate archive based on the predicted objective and measure values.

Note that since DSAGE is independent of the QD algorithm, the *ask* and *tell* methods abstract out the QD algorithm's details. For example, when the QD algorithm is MAP-Elites or CMA-ME, *tell* adds solutions if the cell in the measure space that they belong to is empty or if the existing solution in that cell has a lower objective. For CMA-ME, *tell* also includes updating internal CMA-ES parameters.

The agent simulation phase (lines 21–28) inserts a subset of solutions from the surrogate archive into the ground-truth archive. This phase begins by selecting the subset of solutions from the surrogate archive (line 21). The selected solutions are evaluated by generating the corresponding environment (line 23) and simulating a fixed agent to obtain the true objective and measure values, as well as ancillary agent behavior data (line 24). Evaluation data is appended to the dataset, and solutions that improve their corresponding cell in the ground-truth archive are added to that archive (lines 25, 26).

In the model improvement phase (line 29), the surrogate model is trained in a self-supervised manner through the supervision provided by the agent simulations and the ancillary agent behavior data.

The algorithm is initialized by generating random solutions and simulating the agent in the corresponding environments (lines 2-8). Subsequently, every outer iteration (lines 10-30) consists of model exploitation followed by agent simulation and ending with model improvement.

---

**Algorithm 1:** Deep Surrogate Assisted Generation of Environments (DSAGE)

**Input:** $N$: Maximum number of evaluations, $n_{rand}$: Number of initial random solutions, $N_{exploit}$: Number of iterations in the model exploitation phase, $B$: Batch size for the model exploitation QD optimizer

**Output:** Final version of the ground-truth archive $\mathcal{A}_{gt}$

1  Initialize the ground-truth archive $\mathcal{A}_{gt}$, the dataset $\mathcal{D}$, and the deep surrogate model $sm$
2  $\Theta \leftarrow generate\_random\_solutions(n_{rand})$
3  **for** $\theta \in \Theta$ **do**
4     $env \leftarrow g(\boldsymbol{\theta})$
5     $f, \boldsymbol{m}, \boldsymbol{y} \leftarrow evaluate(env)$
6     $\mathcal{D} \leftarrow \mathcal{D} \cup (\boldsymbol{\theta}, f, \boldsymbol{m}, \boldsymbol{y})$
7     $\mathcal{A}_{gt} \leftarrow add\_solution(\mathcal{A}_{gt}, (\boldsymbol{\theta}, f, \boldsymbol{m}))$
8  **end**
9  $evals \leftarrow n_{rand}$
10 **while** $evals < N$ **do**
11     Initialize a QD optimizer $qd$ with the surrogate archive $\mathcal{A}_{surrogate}$
12     **for** $itr \in \{1, 2, \ldots, N_{exploit}\}$ **do**
13         $\Theta \leftarrow qd.ask(B)$
14         **for** $\theta \in \Theta$ **do**
15             $env \leftarrow g(\boldsymbol{\theta})$
16             $\hat{y} \leftarrow sm.predict\_ancillary(env)$
17             $\hat{f}, \hat{\boldsymbol{m}} \leftarrow sm.predict(env, \hat{y})$
18             $qd.tell(\boldsymbol{\theta}, \hat{f}, \hat{\boldsymbol{m}})$
19         **end**
20     **end**     Model Exploitation
21     $\Theta \leftarrow select\_solutions(\mathcal{A}_{surrogate})$
22     **for** $\theta \in \Theta$ **do**
23         $env \leftarrow g(\boldsymbol{\theta})$
24         $f, \boldsymbol{m}, \boldsymbol{y} \leftarrow evaluate(env)$
25         $\mathcal{D} \leftarrow \mathcal{D} \cup (\boldsymbol{\theta}, f, \boldsymbol{m}, \boldsymbol{y})$     Agent Simulation
26         $\mathcal{A}_{gt} \leftarrow add\_solution(\mathcal{A}_{gt}, (\boldsymbol{\theta}, f, \boldsymbol{m}))$
27         $evals \leftarrow evals + 1$
28     **end**
29     $sm.train(\mathcal{D})$     Model Improvement
30 **end**

---

**Self-supervised prediction of ancillary agent behavior data.** By default, a surrogate model directly predicts the objective and measure values based on the initial state of the environment and the agent (provided in the form of a one-hot encoded image). However, we anticipate that direct prediction will be challenging in some domains, as it requires understanding the agent's trajectory in the environment. Thus, we provide additional supervision to the surrogate model in DSAGE via a two-stage self-supervised process.

First, a deep neural network predicts ancillary agent behavior data. In our work, we obtain this data by recording the expected number of times the agent visits each discretized tile in the environment, resulting in an "occupancy grid." We then concatenate the predicted ancillary information, i.e., the predicted occupancy grid, with the one-hot encoded image of the environment and pass them through another deep neural network to obtain the predicted objective and measure values. We use CNNs for both predictors and include architecture details in Appendix B. As a baseline, we compare our model with a CNN that directly predicts the objective and measure values without the help of ancillary data.

**Downsampling to select solutions from the surrogate archive.** After the model exploitation phase, the surrogate archive is populated with solutions that were predicted to be high-performing and diverse. Hence, a basic selection mechanism (line 21) would select all solutions from the surrogate archive, identical to DSA-ME [19]. However, if the surrogate archive is overly populated, full selection may result in a large number of ground-truth evaluations per outer-loop iteration, leading to fewer outer loops and less surrogate model training. To balance the trade-off between evaluating solutions from the surrogate archive and training the surrogate model, we only select a subset of solutions for evaluation by downsampling the surrogate archive. Downsampling uniformly divides the surrogate archive into sub-regions of cells and selects a random solution from each area.

## 5 Domains

We test our algorithms in two benchmark domains from prior work: a Maze domain [82, 3, 4] with a trained ACCEL agent [4] and a Mario domain [83, 16] with an A* agent [22]. We select these domains because, despite their relative simplicity (each environment is represented as a 2D grid of tiles), agents in these environments exhibit complex and diverse behaviors.

In the Maze domain, we directly search for different mazes, with the QD algorithm returning the layout of the maze. In the Mario domain, we search for latent codes that are passed through a pre-trained GAN, similar to the corresponding previous work.

We select the objective and measure functions as described below. Since the agent or the environment dynamics are stochastic in each domain, we average the objective and measure values over 50 episodes in the Maze domain and 5 episodes in the Mario domain.

**Maze.** We set a binary objective function $f$ that is 1 if the generated environment is solvable and 0 otherwise, indicating the validity of the environment. Since we wish to generate visually diverse levels that offer a range of difficulty level for the agent, we select as measures (1) *number of wall cells* (range: $[0, 256]$), and (2) *mean agent path length* (range: $[0, 648]$, where 648 indicates a failure to reach the goal).

**Mario.** Since we wish to generate playable levels, we set the objective as the *completion rate*, i.e., the proportion of the level that the agent completes before dying. We additionally want to generate environments that result in qualitatively different agent behaviors, thus we selected as measures: (1) *sky tiles*, the number of tiles of a certain type that are in the top half of the 2D grid (range: $[0, 150]$), (2) *number of jumps*, the number of times that the A* agent jumps during its execution (range: $[0, 100]$).

See Appendix A for further environment details.

## 6 Experiments

### 6.1 Experiment Design

**Independent variables.** In each domain (Maze and Mario), we follow a between-groups design, where the independent variable is the algorithm. We test the following algorithms:

*DSAGE*: The proposed algorithm that includes predicting ancillary agent behavior data and downsampling the surrogate archive (Sec. 4).

*DSAGE-Only Anc*: The proposed algorithm with ancillary data prediction and no downsampling, i.e., selecting all solutions from the surrogate archive.

*DSAGE-Only Down*: The proposed algorithm with downsampling and no ancillary data prediction.

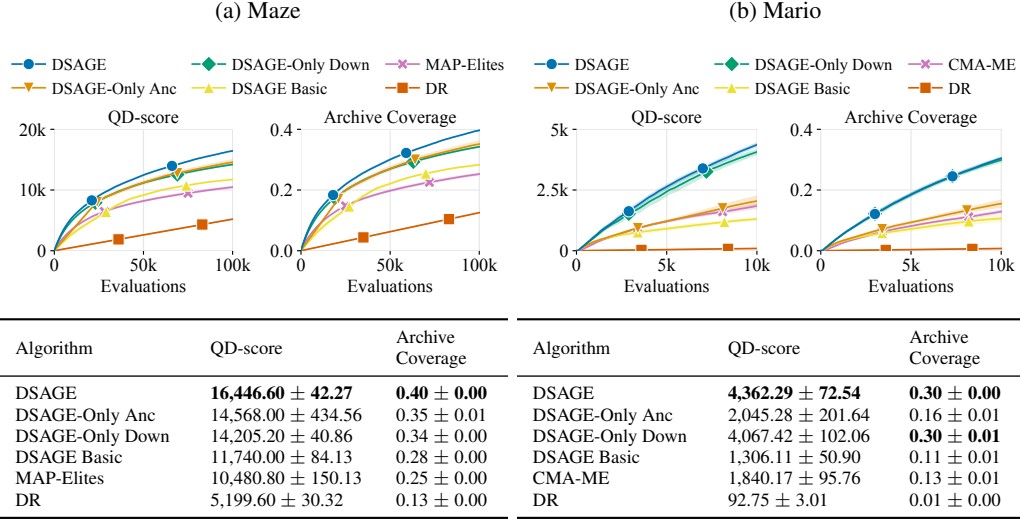

Figure 2: QD-score and archive coverage attained by baseline QD algorithms and DSAGE in the Maze and Mario domains over 5 trials. Tables and plots show mean and standard error of the mean.

*DSAGE Basic*: The basic version of the proposed algorithm that selects all solutions from the surrogate archive and does not predict ancillary data.

*Baseline QD*: The QD algorithm without surrogate assistance. We follow previous work [16] and use CMA-ME for the Mario domain. Since CMA-ME operates only in continuous spaces, we use MAP-Elites in the discrete Maze domain.

*Domain Randomization (DR) [84–86]*: Algorithm that generates and evaluates random solutions, i.e., wall locations in the maze domain and the latent code to pass through the GAN in the Mario domain.

**Dependent variables.** We measure the quality and diversity of the solutions with the QD-score metric [26](Eq. 1). As an additional metric of diversity, we also report the archive coverage. We run each algorithm for 5 trials in each domain.

**Hypothesis.** *We hypothesize that DSAGE will result in a better QD-score than DSAGE Basic in all domains, which in turn will result in better performance than the baseline QD algorithm. DSAGE, DSAGE Basic, and the baseline QD algorithm will all exceed DR.* We base this hypothesis on previous work [25, 17] which shows that QD algorithms outperform random sampling in a variety of domains, as well as previous work [18, 19] which shows that surrogate-assisted MAP-Elites outperforms standard MAP-Elites in design optimization and Hearthstone domains. Furthermore, we expect that the additional supervision through ancillary agent behavior data and downsampling will result in DSAGE performing significantly better than DSAGE Basic.

## 6.2 Analysis

Fig. 2 summarizes the results obtained by the six algorithms on the Maze and the Mario domains.

One-way ANOVA tests showed a significant effect of the algorithm on the QD-score for the Maze ($F(5, 24) = 430.98, p < 0.001$) and Mario ($F(5, 24) = 238.09, p < 0.001$) domains.

Post-hoc pairwise comparisons with Bonferroni corrections showed that DSAGE outperformed DSAGE Basic, Baseline QD, and DR in both the Maze and the Mario domains ($p < 0.001$). Additionally, DSAGE Basic outperformed MAP-Elites and DR in the Maze domain ($p < 0.001$), while it performed significantly worse than the QD counterpart, CMA-ME, in the Mario domain ($p = 0.003$). Finally, Baseline QD outperformed DR in both the Maze and Mario domains ($p < 0.001$).

These results show that deep surrogate assisted generation of environments results in significant improvements compared to quality diversity algorithms without surrogate assistance. They also show that adding ancillary agent behavior data and downsampling is important in both domains. Without these components, DSAGE Basic has limited or no improvement compared to the QD algorithm

Table 1: Number of evaluations required to reach a QD-score of 10480.8 in the Maze domain and 1306.11 in the Mario domain.

(a) Maze

| Algorithm | Evaluations |
|---|---|
| DSAGE | **33,930.40 $\pm$ 1,411.04** |
| DSAGE-Only Anc | 51,919.60 $\pm$ 8,254.24 |
| DSAGE-Only Down | 42,816.60 $\pm$ 691.38 |
| DSAGE Basic | 85,328.60 $\pm$ 2,947.24 |
| MAP-Elites | 100,000 |

(b) Mario

| Algorithm | Evaluations |
|---|---|
| DSAGE | **2,464.40 $\pm$ 356.36** |
| DSAGE-Only Anc | 7,727.40 $\pm$ 1,433.33 |
| DSAGE-Only Down | **2,768.60 $\pm$ 586.34** |
| DSAGE Basic | 10,000 |
| CMA-ME | 5,760.00 $\pm$ 516.14 |

Table 2: Mean absolute error of the objective and measure predictions by the surrogate models.

| | Maze | | | Mario | | |
|---|---|---|---|---|---|---|
| Algorithm | Objective MAE | Number of Wall Cells MAE | Mean Agent Path Length MAE | Objective MAE | Number of Sky Tiles MAE | Number of Jumps MAE |
| DSAGE | 0.03 | 0.37 | 96.58 | 0.10 | 1.10 | 7.16 |
| DSAGE-Only Anc | 0.04 | 0.96 | 95.14 | 0.20 | 1.11 | 9.97 |
| DSAGE-Only Down | 0.10 | 0.95 | 151.50 | 0.11 | 0.87 | 6.52 |
| DSAGE Basic | 0.18 | 5.48 | 157.69 | 0.20 | 2.16 | 10.71 |

without surrogate assistance. Additionally, domain randomization is significantly worse than DSAGE as well as the baselines. The archive coverage and consequently the QD-score is negligible in the Mario domain since randomly sampled latent codes led to little diversity in the levels.

Table 1 shows another metric of the speed-up provided by DSAGE: the number of evaluations (agent simulations) required to reach a fixed QD-score. We set this fixed QD-score to be 10480.8 in the Maze domain and 1306.11 in the Mario domain, which are the mean QD-scores of MAP-Elites and DSAGE Basic, respectively, in those domains. DSAGE reaches these QD-scores faster than the baselines do.

To assess the quality of the trained surrogate model, we create a combined dataset consisting of data from one run of each surrogate assisted algorithm. We use this dataset to evaluate the surrogate models trained from separate runs of DSAGE and its variants. Table 2 shows the mean absolute error (MAE) of the predictions by the surrogate models. The model learned by DSAGE Basic fails to predict the agent-based measures well. It has an MAE of 157.69 for the mean agent path length in Maze and MAE = 10.71 for the number of jumps in Mario. In contrast, the model learned by DSAGE makes more accurate predictions, with MAE = 96.58 for mean agent path length and MAE = 7.16 for number of jumps. We provide detailed results of the surrogate model predictions in Appendix B.1.

## 6.3 Ablation Study

Sec. 4 describes two key components of DSAGE: (1) self-supervised prediction of ancillary agent behavior data, and (2) downsampling to select solutions from the surrogate archive. We perform an ablation study by treating the inclusion of ancillary data prediction (ancillary data / no ancillary data) and the method of selecting solutions from the surrogate archive (downsampling / full selection) as independent variables. A two-way ANOVA for each domain showed no significant interaction effects. We perform a main effects analysis for each independent variable.

**Inclusion of ancillary data prediction.** A main effects analysis for the inclusion of ancillary data prediction showed that algorithms that predict ancillary agent behavior data (DSAGE, DSAGE-Only Anc) performed significantly better than their counterparts with no ancillary data prediction (DSAGE-Only Down, DSAGE Basic) in both domains ($p < 0.001$).

Fig. 2 shows that predicting ancillary agent behavior data also resulted in a larger mean coverage for Maze, while it has little or no improvement for Mario. Additionally, as shown in Table 2, predicting ancillary agent behavior data helped improve the prediction of the mean agent path length in the Maze domain but provided little improvement to the prediction of the number of jumps in the Mario domain. The reason is that in the Maze domain, the mean agent path length is a scaled version of the sum of the agent's tile occupancy frequency, hence the two-stage process which predicts the occupancy grid first is essential for improving the accuracy of the model. On the other hand, the

presence of a jump in Mario depends not only on cell occupancy, but also on the structure of the level and the sequence of the occupied cells.

**Method of selecting solutions from the surrogate archive.** A main effects analysis for the method of selecting solutions from the surrogate archive showed that the algorithms with downsampling (DSAGE, DSAGE-Only Down) performed significantly better than their counterparts with no downsampling (DSAGE-Only Anc, DSAGE Basic) in both domains ($p < 0.001$).

A major advantage of downsampling is that it decreases the number of ground-truth evaluations in each outer iteration. Thus, for a fixed evaluation budget, downsampling results in a greater number of outer iterations. For instance, in the Maze domain, runs without downsampling had only 6-7 outer iterations, while runs with downsampling had approximately 220 outer iterations. More outer iterations leads to more training and thus higher accuracy of the surrogate model. In turn, a more accurate surrogate model will generate a better surrogate archive in the inner loop.

We include an ablation in Appendix D to test between two possible explanations for why having more outer iterations helps with performance: (1) larger number of training epochs, (2) more updates to the dataset allowing the surrogate model to iteratively correct its own errors. We observed that iterative correction accounted for most of the performance increase with downsampling.

The second advantage of downsampling is that it selects solutions evenly from all regions of the measure space, thus creating a more balanced dataset. This helps train the surrogate model in parts of the measure space that are not frequently visited. We include an additional baseline in Appendix E in which we select a subset of solutions uniformly at random from the surrogate archive instead of downsampling. We observe that downsampling has a slight advantage over uniform random sampling in the Maze domain.

Furthermore, if instead of downsampling we sampled multiple solutions from nearby regions of the surrogate archive, the prediction errors could cause the solutions to collapse to a single cell in the ground-truth archive, resulting in many solutions being discarded.

Overall, our ablation study shows that both predicting the occupancy grid as ancillary data and downsampling the surrogate archive independently help improve the performance of DSAGE.

### 6.4  Qualitative Results

Fig. 3 and Fig. 4 show example environments generated by DSAGE in the Maze and Mario domains.

Having the mean agent path length as a measure in the Maze domain results in environments of varying difficulty for the ACCEL agent. For instance, we observe that the environment in Fig. 3(a) has very few walls, yet the ACCEL agent gets stuck in the top half of the maze and is unable to find the goal within the allotted time. On the other hand, the environment in Fig. 3(d) is cluttered and there are multiple dead-ends, yet the ACCEL agent is able to reach the goal.

Fig. 4 shows that the generated environments result in qualitatively diverse behaviors for the Mario agent too. Level (b) only has a few sky tiles and is mostly flat, resulting in a small number of jumps. Level (c) has a "staircase trap" on the right side, forcing the agent to perform continuous jumps to escape and complete the level. We include videos of the playthroughs in the supplemental material.

## 7  Societal Impacts

By introducing surrogate models into quality diversity algorithms, we can efficiently generate environments that result in diverse agent behaviors. While we focused on an RL agent in a Maze domain and a symbolic agent in a Mario game domain, our method can be applied to a variety of agents and domains. This can help with testing the robustness of agents, attaining insights about their behavior, and discovering edge cases before real-world deployment. Furthermore, we anticipate that in the future, closing the loop between environment generation and agent training can improve the ability of agents to generalize to new settings and thus increase their widespread use.

Our work may also have negative impacts. Training agents in diverse environments can be considered as a step towards open-ended evolution [87], which raises concerns about the predictability and safety of the emergent agent behaviors [88, 89]. Discovering corner cases that result in unwanted behaviors or catastrophic failures may also be used maliciously to reveal vulnerabilities in deployed agents [90].

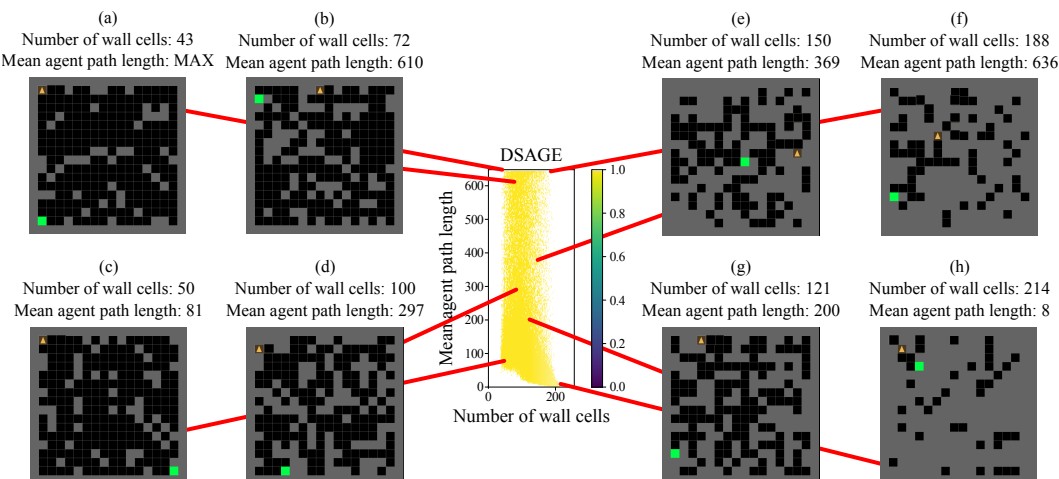

Figure 3: Archive and levels generated by DSAGE in the Maze domain. The agent's initial position is shown as an orange triangle, while the goal is a green square.

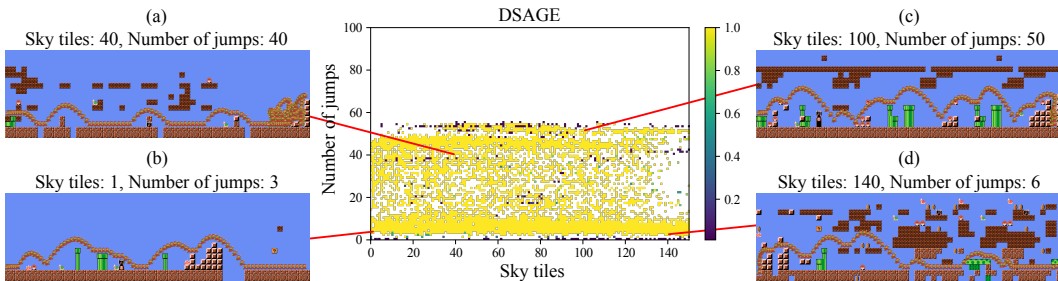

Figure 4: Archive and levels generated by DSAGE in the Mario domain. Each level shows the path Mario takes, starting on the left of the level and finishing on the right.

## 8    Limitations and Future Work

Automatic environment generation is a rapidly growing research area with a wide range of applications, including designing video game levels [46, 53, 54], training and testing autonomous agents [74, 76, 1, 3, 4], and discovering failures in human-robot interaction [17, 16]. We introduce the DSAGE algorithm, which efficiently generates a diverse collection of environments via deep surrogate models of agent behavior.

Our paper has several limitations. First, occupancy grid prediction does not encode temporal information about the agent. While this prediction allows us to avoid the compounding error problem of model-based RL [91], forgoing temporal information makes it harder to predict some behaviors, such as the number of jumps in Mario. We will explore this trade-off in future work.

Furthermore, we have studied 2D domains where a single ground-truth evaluation lasts between a few seconds and a few minutes. We are excited about the use of surrogate models to predict the performance of agents in more complex domains with expensive, high-fidelity simulations [92].

## Acknowledgments and Disclosure of Funding

This work was partially funded by NSF CAREER (#2145077) and NSF GRFP (#DGE-1842487). One of the GPUs used in the experiments was awarded by the NVIDIA Academic Hardware Grant. We thank J. Parker-Holder et al., the authors of the ACCEL agent, for providing a pre-trained model of the agent for our experiments. We also thank Ya-Chuan Hsu for providing invaluable feedback.

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
