# OpenReview forum: "Deep Surrogate Assisted Generation of Environments"
_NeurIPS.cc/2022/Conference — NeurIPS 2022 Accept_

### Official Review · Reviewer_KEnX · 2022-07-05

**Rating:** 7
**Confidence:** 4
**Soundness:** 4 excellent
**Presentation:** 4 excellent
**Contribution:** 3 good

**Summary:**

This paper presents DSAGE, a new approach for generating environments that elicit diverse behaviors for a given agent. The authors consider two settings: 1) a maze environment using an RL agent and 2) Mario with an A* planner. In both cases, DSAGE does exactly as intended. In a set of comprehensive experiments the authors show that each individual component in DSAGE plays an important role, which will be helpful for future work. This paper could have a high impact as an approach for evaluating the robustness for RL agents, and thus I believe it is a useful contribution and should be presented at NeurIPS.

**Questions:**

1. The environments considered are both 2D navigation tasks where DSAGE is essentially “landscaping”. Does this approach work for other settings such as continuous control of robotic environments?
2. Hate to ask for experiments in the rebuttal, but, is it possible to include a random environment generation baseline (i.e. just Domain Randomization)? I would be curious to see how much variation in behavior occurs just using random samples.

**Limitations:**

Limitations are discussed in Section 8. This meets the requirement.

**Strengths And Weaknesses:**

\+ The environments generated by DSAGE are qualitatively diverse, and watching the videos there are clearly some interesting settings that break both of the agents. This provides us with a greater understanding of their respective robustness.

\+ The experimental design is rigorous with many baselines and ablations.

\+ The work is on an important topic that is currently receiving significant interest. This work could have an elevated impact as a result.

\- I was hoping to see some sort of "summary" of the coverage/success of each agent with different regions of failure. For example, some way to analyze how much of the total environment space can be solved by an agent and what types of environments it typically does well/poorly on. It would be super interesting to then use this as a means to compare agents in the same environment. For example, how does ACCEL compare to other RL agents/planners such as PAIRED or A*? If DSAGE is capable of producing a summary like this, then that would increase its impact and I would move to a stronger accept.

Minor:
- I think the citation for MiniGrid is missing, which is the environment for the Maze experiments.

---

> ### Author Response · Authors · 2022-08-02
> **Response to Reviewer KEnX**
>
> > I was hoping to see some sort of "summary" of the coverage/success ... move to a stronger accept.
>
> We agree that a performance summary for different agents would be interesting and impactful. We decided to focus the contribution of this paper on the DSAGE algorithm and assess with ablation studies the performance of DSAGE compared to baselines. We believe that DSAGE is a powerful framework that allows benchmarking different types of agents in different environments. While the heatmaps such as those in Fig. 11-15 in the Appendix provide intuition on the variety of behaviors of one type of agent, a fair comparison of the behavior of different types of agents would require an extensive analysis that we believe would be a studies paper on its own. We would also like to note that while we had access to pre-trained ACCEL agents for the Maze domain, the full training pipeline for ACCEL and PLR was unavailable at the time of submission, and thus, we would be unable to train those agents for the Mario domain. In the future, we plan to explore a study paper where we use DSAGE to qualitatively and quantitatively compare the behavior of different types of agents in different environments.
>
> > I think the citation for MiniGrid is missing, which is the environment for the Maze experiments.
>
> Thank you for pointing this out. The citation for Minigrid is currently in Appendix A, but we have moved it to Sec. 5 in our revised paper.
>
> > The environments considered are both 2D navigation tasks where DSAGE is essentially “landscaping”. Does this approach work for other settings such as continuous control of robotic environments?
>
> The reviewer correctly points out that we evaluate DSAGE on 2D domains. We selected these domains because these are the types of environments used in previous work on generalizable RL agents, e.g., PAIRED, ACCEL. However, we would like to point out that environment generation with QD – without the use of surrogate models – has already been applied in 3D robotics domains (see “Evaluating human-robot interaction algorithms in shared autonomy via quality diversity scenario generation”, Fontaine and Nikolaidis 2021).
>
> We note that while the proposed surrogate model focuses only on 2D domains, there has been exciting recent work on trajectory predictions for 3D scenes (see “Transformer Networks for Trajectory Forecasting” by Giuliari et al.). Future work will explore integrating these predictive models as surrogate models in our framework. We believe that as RL agents become more and more capable and are shown to operate robustly in more complex domains, there will be an increasing need to verify their robustness, and we view our framework as a step in that direction.
>
> > ... is it possible to include a random environment generation baseline (i.e. just Domain Randomization)? I would be curious to see how much variation in behavior occurs just using random samples.
>
> Following the suggestion by the reviewer, we ran the Maze and the Mario experiments with domain randomization. The generation pipeline was kept the same as described in Appendix A, but the solutions were generated randomly instead of through a QD optimizer. In Maze, the wall tiles were selected uniformly randomly while in Mario, the latent code input to the GAN was sampled from a Gaussian with mean 0 and standard deviation 0.2 (the same parameters used to generate the initial set of solutions in DSAGE).
>
> The QD score and the archive coverage for domain randomization were as follows:
>
> Maze domain:
>
> | Algorithm            | QD-score               | Archive Coverage |
> |----------------------|------------------------|------------------|
> | Domain Randomization | 5199.60 $\pm$ 30.32    | 0.13 $\pm$ 0.00  |
> | DSAGE                | 16,446.60 $\pm$ 42.27  | 0.40 $\pm$ 0.00  |
> | DSAGE-Only Anc       | 14,568.00 $\pm$ 434.56 | 0.35 $\pm$ 0.01  |
> | DSAGE-Only Down      | 14,205.20 $\pm$ 40.86  | 0.34 $\pm$ 0.00  |
> | DSAGE Basic          | 11,740.00 $\pm$ 84.13  | 0.28 $\pm$ 0.00  |
> | MAP-Elites           | 10,480.80 $\pm$ 150.13 | 0.25 $\pm$ 0.00  |
>
> Mario domain:
>
> | Algorithm            | QD-score              | Archive Coverage |
> |----------------------|-----------------------|------------------|
> | Domain Randomization | 92.75 $\pm$ 3.01      | 0.01 $\pm$ 0.00  |
> | DSAGE                | 4,362.29 $\pm$ 72.54  | 0.30 $\pm$ 0.00  |
> | DSAGE-Only Anc       | 2,045.28 $\pm$ 201.64 | 0.16 $\pm$ 0.01  |
> | DSAGE-Only Down      | 4,067.42 $\pm$ 102.06 | 0.30 $\pm$ 0.01  |
> | DSAGE Basic          | 1,306.11 $\pm$ 50.90  | 0.11 $\pm$ 0.01  |
> | CMA-ME               | 1,840.17 $\pm$ 95.76  | 0.13 $\pm$ 0.01  |
>
> Hence, domain randomization is much worse than DSAGE as well as the baselines. We think the extremely low score in the Mario domain is due to the latent codes sampled from a Gaussian not leading to a diverse set of environments. We have included these results in Sec. 6.2 of our revised paper, and we thank the reviewer for their suggestion.

---

> > ### Comment · Reviewer_KEnX · 2022-08-04
> > **Thank you for your response :)**
> >
> > I think the new DR baseline is interesting as it grounds the rest of the results, so thank you for adding it.
> >
> > To me this is a solid contribution in an interesting/frontier area of research. The paper should be accepted so I am raising my score.

---

> > > ### Author Response · Authors · 2022-08-04
> > > **Thank you**
> > >
> > > Thank you for the detailed comments and suggestions. We appreciate your score increase.

---

### Official Review · Reviewer_PfHr · 2022-07-10

**Rating:** 7
**Confidence:** 4
**Soundness:** 3 good
**Presentation:** 3 good
**Contribution:** 3 good

**Summary:**

This paper accelerates the quality diversity procedure for environment generation by using a learned predictor rather than agent evaluations. They add auxiliary losses on occupancy measures to improve the prediction process and a downsampling procedure to decrease the number of candidates.

**Questions:**

- The agents are stochastic, why are the surrogate predictors deterministic? Would some of the challenges in prediction go away with a stochastic predictor?
- Why would downsampling help with prediction accuracy as it appears to?

**Limitations:**

The authors adequately address the limitations.

**Strengths And Weaknesses:**

### Strengths:
- This paper appears to be a real accelerator for QD algorithms and the authors carefully demonstrate that each component of the algorithm contributes to the improvement. The algorithm seems very useful.
- The paper is very clearly written.

### Weaknesses:
- The paper is quite similar to a related work and the addition of an auxiliary prediction loss is not a huge novelty.

### A few writing comments:
- In line 138 I’m not sure that “ancillary agent behavior” is a phrase whose meaning is universally known and I’m not sure it was defined before this point? It’s explained in 152 but when I read it on 138 I didn’t understand at the moment and was just confused instead.
- Perhaps I missed it but I didn’t see anywhere where $y$ was defined. I understand that it’s the ancillary but I don’t think that fact is in the paper except in algorithm 1.
- The CMA-ES step seems to be missing from algorithm 1?
- This is a personal preference but given that the method performing better than baselines is the point of the paper, the inclusion of the “hypothesis” in line 222 seems very unnecessary.
- I think either Figure 7 or Table 1 from the supplement should be in the main paper, since improved prediction is one of the main contributions of the paper
- The method seems similar to “Deep surrogate assisted MAP-Elites for automated hearthstone deckbuilding” which is acknowledged in the related work. There the differentiating claim is that that work applies prediction to the decks rather than the agent performance. In that case, it seems the differentiating factor of this work is a series of tools that enable these methods to work in the agent-based case (namely the ancillary information and the downsampling helping with prediction) rather than a new method. This is not super clear from the paper as it seems DSAGE-BASIC is essentially the method from the hearthstone paper? If these are the main differentiating factors, then more analysis of the impact of these tools should probably be in the main paper rather than in the supplement and claiming that a new algorithm has been constructed seems unnecessary. For example, that downsampling improved run-time by 4x is pretty neat and the speedup of the surrogate loop is worth mentioning!

---

> ### Author Response · Authors · 2022-08-02
> **Response to Reviewer PfHr (part 1/2)**
>
> > In line 138 I’m not sure that “ancillary agent behavior” is a phrase whose meaning is universally known and I’m not sure it was defined before this point? It’s explained in 152 but when I read it on 138 I didn’t understand at the moment and was just confused instead.
>
> We agree that “ancillary agent behavior data” is not a well-known term — we created this term to describe the additional data that our surrogate model is predicting. In our case, this data consists of the occupancy grid for the maze and Mario agents. We have clarified this term in Sec. 4 of our revised paper.
>
> > Perhaps I missed it but I didn’t see anywhere where y was defined. I understand that it’s the ancillary but I don’t think that fact is in the paper except in algorithm 1.
>
> Thank you for pointing this out. y does indeed refer to the ancillary agent behavior data. We have clarified this in Sec. 4 of our revised paper.
>
> > The CMA-ES step seems to be missing from algorithm 1?
>
> Algorithm 1 does not include the CMA-ES step because we abstract the QD optimizer into a subroutine which we call to generate solutions when searching/exploiting the surrogate model. We wanted to write Algorithm 1 in a way that makes it independent of the actual QD algorithm being implemented. Thus, instead of tying ourselves to a specific QD optimizer, such as CMA-ME, Algorithm 1 refers to the QD optimizer simply as “qd.” Then, “qd.ask()” and “qd.tell()” handle any necessary internal details of the QD optimizer. We have clarified the roles of the ask and tell methods in Sec. 4 of our revised paper.
>
> Specifically for CMA-ME, qd.tell() includes updating the CMA-ES parameters and adding the solution to the archive. Meanwhile, for MAP-Elites, qd.tell() adds the solution to the archive.
>
> > This is a personal preference but given that the method performing better than baselines is the point of the paper, the inclusion of the “hypothesis” in line 222 seems very unnecessary.
>
> We agree with the reviewer that the hypothesis reiterates the main goal of the paper. Our motivation for explicitly writing the hypothesis was to make clear the aims of the experiments, which guide our statistical analysis in section 6.2.
>
> > I think either Figure 7 or Table 1 from the supplement should be in the main paper, since improved prediction is one of the main contributions of the paper
>
> We agree with the reviewer, and we have added Table 1 (now Table 2) and a summary of the discussion in Appendix B.1 to Sec. 6.2 and Sec. 6.3 of our revised paper.
>
> > The method seems similar to “Deep surrogate assisted MAP-Elites for automated hearthstone deckbuilding” ... the speedup of the surrogate loop is worth mentioning!
>
> The reviewer correctly points out that our work builds on DSA-ME (Deep Surrogate Assisted MAP-Elites). However, we would like to note that DSA-ME was tested only on one domain: generating decks for the Hearthstone game. On the other hand, DSAGE focuses on generating diverse environments – rather than Hearthstone decks – and evaluates these environments with generalizable agents, while using a surrogate model to predict agent performance. To our knowledge, this is the first work in environment generation that uses predictive models to predict agent performance.
>
> We would like to clarify the difference between DSA-ME and our baseline DSAGE Basic. DSA-ME is specific to MAP-Elites only. On the other hand, DSAGE Basic has a different surrogate model that predicts agent performance and can run any QD algorithm as a subroutine. Furthermore, DSAGE Basic allows the integration of generative models to generate environments that match a provided dataset, as in the Mario domain, rather than being restricted to direct search as in DSA-ME. We have clarified the differences between DSAGE Basic and DSA-ME in Sec. 3 of our revised paper.
>
> Nevertheless, we observe that DSAGE Basic performs poorly in both Mario and Maze domains. In the Mario domain, it performs significantly worse than even the baseline QD algorithm, CMA-ME, which does not use a surrogate model!
>
> Thus, as the reviewer correctly points out, the contribution of this work also includes the methods that make DSAGE work well in this application domain, i.e. downsampling and predicting ancillary agent behavior data. We agree that the results, especially the benefit of downsampling, are worth further discussion beyond the ablations in Section 6.3 in the main paper. We have moved Table 1 (now Table 2) and a summary of the discussion in Appendix B.1 to Sec. 6.2 and Sec. 6.3 in our revised paper.

---

> > ### Comment · Reviewer_PfHr · 2022-08-06
> > **Thanks!**
> >
> > Dear authors,
> >
> > Thank you for the response to my questions. I think the changes have made the paper better; I believe the paper should be accepted and I think my score reflects that.

---

> > > ### Author Response · Authors · 2022-08-06
> > > **Thank you**
> > >
> > > Thank you for the detailed review and feedback.

---

> ### Author Response · Authors · 2022-08-02
> **Response to Reviewer PfHr (part 2/2)**
>
> > The agents are stochastic, why are the surrogate predictors deterministic? Would some of the challenges in prediction go away with a stochastic predictor?
>
> The main motivation behind deterministic predictions is that the surrogate predictors are integrated with a QD algorithm. When searching stochastic domains, the most common approach in evolutionary optimization is “explicit averaging” (“Evolutionary optimization in uncertain environments-a survey” by Jin and Branke), which does multiple rollouts and uses the sample mean as an estimate of the objective and measures. Thus, the surrogate model only needs to output a deterministic estimate of the mean to interface with the QD algorithm.
>
> One interesting direction for future work would be to predict both the mean and the variance of the distribution and interface this with current or future QD algorithms that are optimized for noisy domains, e.g., “Fast and stable MAP-Elites in noisy domains using deep grids“ by Flageat and Cully.
>
> > Why would downsampling help with prediction accuracy as it appears to?
>
> The reviewer correctly points out that in Table 1 (Table 2 in the revised version) in the supplemental material, DSAGE tends to have a more accurate surrogate model than DSAGE-Only Anc, and DSAGE-Only Down tends to have a more accurate surrogate model than DSAGE Basic.
>
> The primary reason is that having a smaller surrogate archive for the same evaluation budget results in a larger number of outer loop iterations. There are two possible explanations for why having more outer loop iterations helps with accuracy: One explanation is that the larger number of training epochs, resulting from training the model in each outer iteration, itself helps with accuracy, as in "Grokking: Generalization beyond overfitting on small algorithmic datasets.", by Power et al. The second explanation is based on the fact that at the beginning of training, the surrogate model is inaccurate, and hence, the data generated by evaluating solutions in the surrogate archive would have been incorrectly predicted by the surrogate model. A larger number of outer loop iterations results in a larger number of times the algorithm updates the dataset with these adversarial examples, allowing the surrogate model to iteratively correct its own errors.
>
> We will run an additional ablation in the revised version of the paper, where we increase the number of training epochs for the algorithms that do not use downsampling (DSAGE-Only Anc and DSAGE Basic), making the total number of training epochs the same as that with downsampling. This will allow us to disambiguate the two explanations. We thank the reviewer for pointing this out.
>
> Preliminary results reported in the table below show that the performance of DSAGE Basic and DSAGE-Only Anc with a larger number of training epochs lies in between their original performance and the performance of their counterparts with downsampling (DSAGE-Only Down, DSAGE).
>
> Maze domain:
>
> | Algorithm                    | QD-score               | Archive Coverage |
> |------------------------------|------------------------|------------------|
> | DSAGE                        | 16,446.60 $\pm$ 42.27  | 0.40 $\pm$ 0.00  |
> | DSAGE-Only Anc (more epochs) | 15864                  | 0.38             |
> | DSAGE-Only Anc               | 14,568.00 $\pm$ 434.56 | 0.35 $\pm$ 0.01  |
> | DSAGE-Only Down              | 14,205.20 $\pm$ 40.86  | 0.34 $\pm$ 0.00  |
> | DSAGE Basic (more epochs)    | 12568                  | 0.30             |
> | DSAGE Basic                  | 11,740.00 $\pm$ 84.13  | 0.28 $\pm$ 0.00  |
>
> Mario domain:
>
> | Algorithm                    | QD-score              | Archive Coverage |
> |------------------------------|-----------------------|------------------|
> | DSAGE                        | 4,362.29 $\pm$ 72.54  | 0.30 $\pm$ 0.00  |
> | DSAGE-Only Anc (more epochs) | 2442.63               | 0.18             |
> | DSAGE-Only Anc               | 2,045.28 $\pm$ 201.64 | 0.16 $\pm$ 0.01  |
> | DSAGE-Only Down              | 4,067.42 $\pm$ 102.06 | 0.30 $\pm$ 0.01  |
> | DSAGE Basic (more epochs)    | 1343.65               | 0.10             |
> | DSAGE Basic                  | 1,306.11 $\pm$ 50.90  | 0.11 $\pm$ 0.01  |
>
> Hence, a larger number of training epochs partially accounts for the performance increase and we believe that the iterative correction accounts for the rest. We will state this result in Sec. 6.3 of the revised version of the paper and will add this table to the appendix once we have results from multiple repetitions of these runs.

---

### Official Review · Reviewer_6K7J · 2022-07-11

**Rating:** 3
**Confidence:** 3
**Soundness:** 2 fair
**Presentation:** 2 fair
**Contribution:** 2 fair

**Summary:**

This paper addresses the problem of auto-generating RL environments. Classic environment generation methods could be slow due to expensive and time-consuming simulations. The authors proposed DSAGE, which leverages a surrogate model to predict the behaviors of an agent in new environments. The surrogate model is used to accelerate the generation of environments. The proposed DSAGE is evaluated on two benchmark domains and achieves better QD scores and coverage than the baselines.


**Questions:**

- Could the proposed approach scale to more complex environments beyond 2D grid-based maze and 2D Mario?



- How is the solution quality of the proposed method compared with the classic QD method that needs lots of simulation and long time? The reviewer expected there is a trade-off between solution quality and time.
- Reporting the time needed for each method would be very helpful.
- The symbol $\hat{y}$, $\hat{f}$, $\hat{m}$ are undefined.
- In figure 1, “CMA-ME” and the “Sample latent code” are not explained in the text, which is confusing.
- The term “ancillary agent” is used throughout the paper without formal definition. It confuses the reviewer while reading the paper.
- L 139 reads “The tell method adds the solution to the surrogate archive based on the predicted objective and measure values”. How exactly does the tell method add  solutions? Is there a threshold that is used to determine whether to add a solution or not?
- The selection of the objective f and measures m seem critical. How do you make the selections? In addition, in more complex environments, the number of different measure functions may increase. How does the increased number of measure functions affect the performance of the proposed approach?

**Limitations:**

The authors adequately addressed the limitations and potential negative impact in the paper.



**Strengths And Weaknesses:**

Originality  /  Significance:
Existing works that rely on QD require lots of simulations which are expensive and slow.  The ideal of using a surrogate model to predict the behavior of an agent and thus reduce simulation time is intriguing and reasonable. This is an interesting combination of behavior prediction and QD approaches.

Quality:
The reviewer has some concern regarding the experimental results.

1.The author claims that the proposed DSAGE could accelerate the environment generation process. However, the claim is not supported by experimental results. In the experimental section, no speed information is reported. It would be helpful to compare the speed of the proposed DSAGE with baselines.
2.The author claims that diverse auto-generated environments could improve agent’s capability of generalizing to new environments. The claim is also not supported by experiments.
3. The proposed method is evaluated on two simple domains: grid-based maze and 2D Mario. It is unclear if the proposed approach could salce to more complex domains.

Clarity:
The reviewer is not an expert in environment generation but has expertise in RL. However, the reviewer found the paper somewhat difficult to follow. Improvements may be required. Please see the Question section for details.

---

> ### Author Response · Authors · 2022-08-02
> **Response to Reviewer 6K7J (part 1/3)**
>
> > The author claims that the proposed DSAGE could accelerate the environment generation process. However, the claim is not supported by experimental results. In the experimental section, no speed information is reported. It would be helpful to compare the speed of the proposed DSAGE with baselines.
>
> We note that the speed-up provided by DSAGE in the current experiments is in terms of sample efficiency. We assume that we are allowed a fixed budget of agent simulations and compare the performance of the algorithms after they consume the allowed budget of agent simulations. The x-axis in Fig. 2 shows the number of agent simulations (we have added the axis label to the figure in the revised version).
>
> In Sec. 6.2 of our revised paper, we have reported another metric that might show the speed-up more intuitively: the number of agent simulations required for DSAGE to reach the same QD score as compared to the final QD score of the baselines. The table below shows this metric in both the Maze and the Mario domains:
>
> Number of evaluations (agent simulations) to reach a QD score of 10480.8 in the Maze domain:
>
> | Algorithm       | Evaluations            |
> |:----------------|:-----------------------|
> | DSAGE           | $33930.40 \pm 1411.04$ |
> | DSAGE-Only Anc  | $51919.60 \pm 8254.24$ |
> | DSAGE-Only Down | $42816.60 \pm 691.38$  |
> | DSAGE Basic     | $85328.60 \pm 2947.24$ |
> | MAP-Elites      | $100000$                 |
>
> Number of evaluations (agent simulations) to reach a QD score of 1306.11 in the Mario domain:
>
> | Algorithm       | Evaluations           |
> |:----------------|:----------------------|
> | DSAGE           | $2464.40 \pm 356.36$  |
> | DSAGE-Only Anc  | $7727.40 \pm 1433.33$ |
> | DSAGE-Only Down | $2768.60 \pm 586.34$  |
> | DSAGE Basic     | $10000$                 |
> | CMA-ME          | $5760.00 \pm 516.14$  |
>
> Furthermore, we would like to note that a fair comparison of wall clock times would be difficult since the agent simulation is done on multiple CPUs while the surrogate model training and model exploitation are done on a GPU. Additionally, the episode lengths can vary a lot depending on the generated environment. For example, the maze in Fig. 3h was solved in 8 steps on average, while the one in Fig. 3b took 610 steps. Hence, the wall clock time also depends on the candidate environments generated in the intermediate steps of the algorithm. Formally analyzing this would be beyond the scope of this work which aims to improve the sample efficiency of environment generation via QD algorithms by leveraging a surrogate model.
>
> > The author claims that diverse auto-generated environments could improve agent’s capability of generalizing to new environments. The claim is also not supported by experiments.
>
> We believe that the reviewer refers to the statement in line 302, “... we anticipate that closing the loop between environment generation and agent training can improve the ability of agents to generalize to new settings …”. We would like to underline that this statement is a part of the Societal Impacts section and it is stated as a potential future direction of our algorithm. DSAGE currently provides a method to test the robustness of agents by generating environments, but leveraging the generated environments to improve an agent is an area of future research. We have clarified this further in Sec. 7 of our revised paper.
>
> > The proposed method is evaluated on two simple domains: grid-based maze and 2D Mario. It is unclear if the proposed approach could scale to more complex domains.
>
> Our aim in this paper is to provide a method that efficiently generates environments to test generalizable agents. The current state-of-the-art methods to train such generalizable agents report their results on 2D domains, and hence, we selected two of these domains in this paper. We observe that even these 2D domains result in a wide range of agent behaviors.
>
> While the proposed surrogate model focuses only on 2D domains, there has been exciting recent work on trajectory predictions for 3D scenes (see “Transformer Networks for Trajectory Forecasting” by Giuliari et al.). Future work will explore integrating these predictive models as surrogate models in our framework. We believe that as RL agents become more and more capable and are shown to operate robustly in more complex domains, there will be an increasing need to verify their robustness, and we view our framework as a step in that direction.

---

> ### Author Response · Authors · 2022-08-02
> **Response to Reviewer 6K7J (part 2/3)**
>
> > How is the solution quality of the proposed method compared with the classic QD method that needs lots of simulation and long time? The reviewer expected there is a trade-off between solution quality and time.
>
> When running the same algorithm, we observe that the performance increases monotonically over time, thus longer time indeed results in better performance. Our goal is to compare different algorithms with respect to sample efficiency. Thus, in Fig. 2 in the paper, we fix the total number of evaluations (agent simulations) and we compare the performance of the algorithms after those many evaluations.
>
> We assess the performance of the algorithms with the QD-score metric (see “Quality diversity: A new frontier for evolutionary computation,” by Pugh et al.), which captures both the quality (how high are the objective values) and the diversity (how much of the measure space is covered) of the set of solutions. Here, we show that for the same number of agent simulations, DSAGE performs better than the baseline in both the Maze and Mario domains. In the Maze domain, the quality of a level is determined by the validity of the level, and all algorithms were able to find valid levels, thus the difference in performance was explained by the ability of DSAGE to fill in the archive with diverse solutions. In the Mario domain, solution quality was defined as the completion percentage by the A* agent and DSAGE was able to find more diverse but also higher quality solutions.
>
> We additionally observe that in Fig. 2, DSAGE always has a larger QD-score than all baselines at any point in time, so for any fixed number of evaluations, DSAGE has the better performance.
>
> Alternatively, we can assess sample efficiency by computing the number of evaluations required to reach a given performance level, as shown in the table above. We observe that DSAGE requires fewer evaluations than the baselines to reach that level.
>
> > The symbol \hat{y}, \hat{f}, \hat{m} are undefined.
>
> The symbols denote the predicted ancillary data, objective value, and measure values respectively. We thank the reviewer for their suggestion and have clarified this in Sec. 4 of our revised paper.
>
> > In figure 1, “CMA-ME” and the “Sample latent code” are not explained in the text, which is confusing.
>
> “CMA-ME” is the algorithm “Covariance Matrix Adaptation MAP-Elites” from the paper “Covariance matrix adaptation for the rapid illumination of behavior space” by Fontaine et al. “Sample latent code” refers to sampling a vector that acts as a latent code to the GAN that generates the environment (only applicable to the Mario domain in this paper). In the revised paper, we have cited CMA-ME in the caption of Fig. 1 and renamed the term “latent code” as “solution” to make it more general. We thank the reviewer for pointing this out.
>
> > The term “ancillary agent” is used throughout the paper without formal definition. It confuses the reviewer while reading the paper.
>
> We agree that “ancillary agent behavior data” is not a well-known term — we created this term to describe the additional data that our surrogate model is predicting. In our case, this data consists of the occupancy grid for the maze and Mario agents. We have added a definition of this term in Sec. 4 of our revised paper.
>
> > L 139 reads “The tell method adds the solution to the surrogate archive based on the predicted objective and measure values”. How exactly does the tell method add solutions? Is there a threshold that is used to determine whether to add a solution or not?
>
> The exact mechanism to add solutions depends on the QD optimization method. In MAP-Elites and CMA-ME, solutions are added if the cell in the measure space that they belong to is empty or if the existing solution in that cell has a lower objective value.
>
> Since our algorithm is independent of the QD optimization method, we simply abstract out the solution generation and solution addition parts as qd.ask and qd.tell respectively. This is in line with the API adopted by popular evolutionary algorithms and QD libraries such as pycma (https://github.com/CMA-ES/pycma) and pyribs (https://pyribs.org/). We have further clarified the role of the ask and tell methods in Sec. 4 of our revised paper.

---

> ### Author Response · Authors · 2022-08-02
> **Response to Reviewer 6K7J (part 3/3)**
>
> > The selection of the objective f and measures m seem critical. How do you make the selections?
>
> In quality diversity (QD), the measure functions and the objective function are part of the problem statement. The QD objective is to discover solutions that are high-quality with respect to the objective and diverse with respect to the measure functions. For the problem of testing agents, our goal is to generate valid environments that elicit a diverse range of behaviors. Thus, we wish our objective to be a metric of environment validity so that we generate valid environments. The measure functions specify the dimensions of behavioral diversity that a designer wishes to obtain. Beyond the measure functions proposed in the main paper, we would like to point out that we propose additional measure functions in Appendix F, e.g., percentage of maze explored, which results in maze environments that range from mostly unexplored to extensively explored by the agents. Previous work has also explored a variety of measure functions for the Mario domain, e.g., number of coins collected, (see “Illuminating mario scenes in the latent space of a generative adversarial network." by Fontaine et al.) in addition to the number of jumps and number of sky tiles that we have in the paper.
>
> We would like to point out that there is an increasing number of research works that focus on learning measure functions from the trajectory data of the agents using dimensionality reduction techniques, e.g., “Unsupervised behaviour discovery with quality-diversity optimisation” by Grillotti and Cully. We view these works as complementary to our environment generation framework.
>
> > In addition, in more complex environments, the number of different measure functions may increase. How does the increased number of measure functions affect the performance of the proposed approach?
>
> Increasing the number of measure functions introduces two challenges to our approach. First, we must predict more measures with the surrogate model. This may be addressed relatively easily by adding more outputs to the model.
>
> Second, and more importantly, increasing the number of measure functions would exponentially increase the volume of the measure space that the QD algorithm has to cover. On one hand, we would need a larger number of evaluations both for our approach and for the baselines. On the other hand, we would need to choose a QD algorithm designed to handle the larger measure space. Scaling QD algorithms to larger measure spaces is an active research topic in the QD literature, and we can apply techniques such as those proposed in “Using centroidal voronoi tessellations to scale up the multidimensional archive of phenotypic elites algorithm” by Vassiliades et al., “Quality-diversity meta-evolution: customising behaviour spaces to a meta-objective” by Bossens and Tarapore, and “Relevance-guided unsupervised discovery of abilities with quality-diversity algorithms” by Grillotti and Cully.
>
> Furthermore, it is often the case that not all parts of the measure space are important. Thus, we can decrease the resolution in parts of the measure space that are less important, to guide the search towards more important regions. Previous work has also explored adjusting the resolution of the measure space dynamically based on the evolving distribution of high-performing solutions ("Mapping hearthstone deck spaces through map-elites with sliding boundaries" by Fontaine et al). Overall, our approach uses QD algorithms as sub-routines, and improvements in these algorithms can improve the performance of our framework when dealing with increased numbers of measure functions.

---

> ### Comment · Reviewer_6K7J · 2022-08-09
> **Thanks for the response**
>
> The reviewer thanks the authors’ response and revision. The additional experimental results on runtime and number evaluations are very helpful. Regarding the environments, is the proposed approach applicable to environments like gMiniWoB, which is used in Gur et al.
>
> Gur et al., Environment Generation for Zero-Shot Compositional Reinforcement Learning, NeurIPS 2021

---

> > ### Author Response · Authors · 2022-08-09
> > **Thank you**
> >
> > Thank you for the thorough review.
> >
> > In the method proposed by Gur et al., gMiniWoB environments are generated via a policy conditioned on an initial observation that is sampled from a Gaussian distribution (described in Sec. 5 of that paper). We believe that the environment generation pipeline that we use in the Mario domain can be applied to gMiniWoB environments as well. For Mario environments, the QD optimizer returns solutions that act as the latent vector for a trained GAN, hence generating different environments. Similarly, diverse gMiniWoB environments can be generated by treating the solutions from the QD optimizer as the initial observation for a trained generator policy. We will add a discussion about this in the future work section of the revised version.
> >
> > If the reviewer is satisfied with our response and revised version, would the reviewer kindly consider upgrading their score?

---

### Official Review · Reviewer_LA7A · 2022-07-12

**Rating:** 4
**Confidence:** 3
**Soundness:** 3 good
**Presentation:** 2 fair
**Contribution:** 2 fair

**Summary:**

This paper proposes a sample-efficient QD environment generation algorithm that maintains a deep surrogate model for predicting agent behaviors in new environments. Results in two benchmark domains show that this method outperforms existing QD environment generation algorithms.

**Questions:**

See weakness 2

**Limitations:**

1. Not scalable to some tasks where the measure functions are inconvenient/impossible to define
2. Need a pre-trained agent
3. Need to compare with more baselines, e.g. PAIRED, ACCEL

**Strengths And Weaknesses:**

Strengths:
Clean method with experimental results and ablations to demonstrate the effectiveness of their method compared with QD baselines

Weakness:
1. Unclear presentation:
    a. Figure 1 is confusing with many arrows and no starting point, while the illustration is not intuitive too
    b. Undefined symbols and operations in pseudo-code, e.g. $y$, qd.ask, qd.tell
2. Lack of efficiency analysis: the experimental results only show the evaluation of solutions, but not how many iterations to take in order to get these solutions.

---

> ### Author Response · Authors · 2022-08-02
> **Response to Reviewer LA7A (part 1/2)**
>
> > Unclear presentation: a. Figure 1 is confusing with many arrows and no starting point, while the illustration is not intuitive too
>
> We would like to point out that the three phases — model improvement, model exploitation, and agent simulation, form the core components of the algorithm and thus are highlighted in different colors in the figure. Furthermore, in Algorithm 1, we have highlighted the pseudocode with the same colors to match Fig. 1. In the revised version of our paper, we have clarified this in the caption for Fig. 1.
>
> > b. Undefined symbols and operations in pseudo-code, e.g. y, qd.ask, qd.tell
>
> We have described qd.ask and qd.tell in lines 135 and 139. Since our algorithm is independent of the QD optimization method, we simply abstract out the solution generation and solution addition parts as qd.ask and qd.tell, respectively. This is in line with the API adopted by popular evolutionary algorithms and QD libraries such as pycma (https://github.com/CMA-ES/pycma) and pyribs (https://pyribs.org/). We have clarified the roles of the ask and tell methods and defined y in section 4 (page 4) of our revised paper.
>
> > Lack of efficiency analysis: the experimental results only show the evaluation of solutions, but not how many iterations to take in order to get these solutions
>
> We note that the speed-up provided by DSAGE in the current experiments is in terms of sample efficiency. We assume that we are allowed a fixed budget of agent simulations and compare the performance of the algorithms after they use up the allowed budget of agent simulations. The x-axis in Fig. 2 shows the number of agent simulations (we have added the axis label to the figure in the revised version).
>
> In Sec. 6.2 of our revised paper, we have reported another metric that might show the speed-up more intuitively: the number of agent simulations required for DSAGE to reach the same QD score as compared to the final QD score of the baselines. The table below shows this metric in both the Maze and the Mario domains:
>
> Number of evaluations (agent simulations) to reach a QD score of 10480.8 in the Maze domain:
>
> | Algorithm       | Evaluations            |
> |:----------------|:-----------------------|
> | DSAGE           | $33930.40 \pm 1411.04$ |
> | DSAGE-Only Anc  | $51919.60 \pm 8254.24$ |
> | DSAGE-Only Down | $42816.60 \pm 691.38$  |
> | DSAGE Basic     | $85328.60 \pm 2947.24$ |
> | MAP-Elites      | $100000$               |
>
> Number of evaluations (agent simulations) to reach a QD score of 1306.11 in the Mario domain:
>
> | Algorithm       | Evaluations           |
> |:----------------|:----------------------|
> | DSAGE           | $2464.40 \pm 356.36$  |
> | DSAGE-Only Anc  | $7727.40 \pm 1433.33$ |
> | DSAGE-Only Down | $2768.60 \pm 586.34$  |
> | DSAGE Basic     | $10000$               |
> | CMA-ME          | $5760.00 \pm 516.14$  |
>
> Hence, DSAGE reaches the given QD score faster than the baselines.

---

> ### Author Response · Authors · 2022-08-02
> **Response to Reviewer LA7A (part 2/2)**
>
> > Not scalable to some tasks where the measure functions are inconvenient/impossible to define
>
> In quality diversity (QD), the measure functions and the objective function are part of the problem statement. The QD objective is to discover solutions that are high-quality with respect to the objective and diverse with respect to the measure functions. For the problem of testing agents, our goal is to generate environments that elicit a diverse range of behaviors, and the measure functions specify the dimensions of behavioral diversity that we wish to obtain.
>
> We note that previous work has explored extensively a wide range of measure functions that result in behavioral diversity of agents, e.g., “Robots that can adapt like animals.” by Cully et al. as well as follow-up work on MAP-Elites for agent behaviors.
>
> Even if there is a measure function that is impossible to define, we can directly use the trajectory data of the agents and apply dimensionality reduction techniques, e.g., “Unsupervised behaviour discovery with quality-diversity optimisation” by Grillotti and Cully, to learn relevant measure functions from the data.
>
> > Need a pre-trained agent
>
> The goal of this paper is to generate environments that elicit diverse behaviors in agents, not to train the agents themselves. We note that while DSAGE tests a pre-trained ACCEL agent in the maze domain, it tests a symbolic A* agent in the Mario domain, and it is not dependent on a specific agent architecture.
>
> > Need to compare with more baselines, e.g. PAIRED, ACCEL
>
> The goal of this paper is to generate environments that result in diverse agent behaviors, not to train agents that generalize to new environments, as in PAIRED and ACCEL. We would like to note that environment generation techniques based on quality diversity predate ACCEL and PAIRED (see "Procedural content generation through quality diversity." by Gravina et al.), and also solve a different problem: PAIRED and ACCEL iteratively search for a single environment that maximizes the regret between a pair of RL-based agents, while QD algorithms in environment generation search simultaneously for a diverse collection of environments.
>
> Additionally, in Sec. 6.2 of our revised paper, we have added an additional baseline, “domain randomization,” that randomly generates environments, as suggested by Reviewer KEnX. Our results show that the proposed method, DSAGE, vastly outperforms that baseline.

---

> > ### Comment · Reviewer_LA7A · 2022-08-10
> > **Thank you for your response**
> >
> > My questions are mostly addressed, and the clarity of the presentation is increased after the revision. I updated my score accordingly. However, I'm still half-convinced by the contribution of the proposed method.
> > >  generate environments that elicit diverse behaviors in agents, not to train the agents themselves
> >
> > It's hard to see whether the generated environments can elicit diverse behavior of agents without training agents on top of them. For example, although the number of wall cells and path length varies, the generated environments may not have significant differences from each other in terms of agent performance. In other words, the coverage or QD score of generated environments doesn't matter much to elicit diverse behavior in agents
> >
> > > The goal of this paper is to generate environments that result in diverse agent behaviors, not to train agents that generalize to new environments, as in PAIRED and ACCEL
> >
> > I agree the goals are different, but the baselines and evaluation metrics are not very common in the RL community. It'll be interesting to see
> >
> > * The QD/coverage of PAIRED/ACCEL during the generation process
> > * Compare the performance of (a) the agent trained with PAIRED vs. (b) the agent trained on environments generated by DSAGE on some unseen validation environments
> >
> > In all, I agree this method made a solid contribution on the line of QD optimization-based environment generation, but additional future work needs to be done to make a larger impact or contribution to a broader community.

---

> > > ### Author Response · Authors · 2022-08-10
> > > **Thank you for the response and some additional clarifications**
> > >
> > > Thank you for reading our response. We are happy that the reviewer finds that this method makes a solid contribution in QD optimization-based environment generation.
> > >
> > > We would like to emphasize that our paper is not focused on RL, but on environment generation motivated by eliciting diverse behaviors of different types (planning-based, RL-based, etc.) of agents, where we treat the agent as a black box. As a result, our insights have applications beyond the testing of RL agents; we can test symbolic agents, such as the Mario agent tested in our paper, and more generally procedurally generate diverse content.
> > >
> > > We would like to point out that the generated environments do result in agent behaviors that are diverse with respect to the specified measure functions. For example, in Figure 4, we have environments that differ drastically in the number of jumps for the Mario agent, in Figure 11 in the supplemental material environments that are different on the percentage of the environment being explored by the trained agent (97% in Figure 11e vs 15% in 11b), in Figure 13 environments that are different on the number of repeated visits by the tested agent (604 in Figure 13e vs 0 in 13h). For every figure we have accompanying videos in the supplemental material that visualize the diverse behaviors.
> > >
> > > With respect to comparing with PAIRED or ACCEL for coverage and QD-score, we would like to point out that if we freeze the training of the agent, PAIRED would only create a single environment that maximizes regret. Changing the agent while generating environments differs from our goal of testing with good behavioral coverage a single black-box agent.
> > >
> > > Our method could be incorporated into a system like ACCEL that does the training and environment generation online. However, this would be an interesting, albeit very different, paper focused on training RL agents, which does not fall into the problem definition of QD optimization for environment generation.

---

> > > > ### Comment · Reviewer_LA7A · 2022-08-10
> > > > **Response to the authors**
> > > >
> > > > Thank you for your response.  I appreciate this clarification on the diverse agent behaviors. I also agree that this paper is orthogonal to PAIRED/ACCEL, which generates somewhat infeasible solutions for a trainable agent, while your method aims at generating feasible and diverse environments for a given agent.
> > > >
> > > > > Our method could be incorporated into a system like ACCEL that does the training and environment generation online
> > > >
> > > > Actually, that's what I'm expecting, as the metrics in the current paper are not convincing under this setup. But I agree it's a different problem with QD optimization. In the future, it'll be interesting to see whether QD optimization is indeed helpful in this loop.

---

### Author Response · Authors · 2022-08-02
**Thank you very much for the detailed reviews.**

Thank you very much for your insightful comments on our work. We appreciate that the reviewers underlined the significance of our work for environment generation, especially the recognition that “this paper appears to be a real accelerator for QD algorithms” and “this paper could have a high impact as an approach for evaluating the robustness for RL agents.” We are glad to see that the reviewers believe we have a “clean method with experimental results and ablations to demonstrate the effectiveness of [our] method,” and that “the paper is very clearly written.”

We have responded to each reviewer’s concerns individually and uploaded a revised version of our paper. We look forward to continuing the discussion.

---

### Meta-Review · Area_Chair_vJRj · 2022-08-25

**Recommendation:** Accept
**Confidence:** Less certain

**Metareview:**

The authors describe a method of scaling the benefits of quality diversity (QD) optimization for automatic generation of RL environments, by replacing costly agent evaluation with a learnt surrogate model. They then demonstrate that this method improves agent performance in two settings: a maze environment using RL and a Mario environment using A* planning.

The reviewers agree that the proposed DSAGE algorithm is both novel and technically sound. Where they disagree is on whether QD optimization as a field of research has inherent value for RL research or the broader NeurIPS readership. To summarize each of their respective stances:

R1: "[...] I still agree this method is solid and improves previous QD optimization methods. But I didn't directly see its contribution to a broader community."

R2: "[...] I have some skepticisms about QD methods in general and prefer unsupervised methods that do not require human specifications of what constitutes an interesting environment. However, it seems unfair to hold this paper responsible for general disagreements with QD style methods and instead seems like the criteria should be whether this is an interesting or useful contribution to researchers who either work on environment design or on researchers who work on QD + RL."

R3: "[...] this is a novel problem setting for the NeurIPS community, combining level generation (from the games community) and RL. It is highly relevant in combination with new work on generalization in RL in particular, and evaluating the robustness of our agents."

I have no great insight on whether QD methods will prove to be valuable for improving agent generalization or robustness in future. But I tend to agree with R3; in my opinion NeurIPS is the right community to be exploring these questions, and this work has made meaningful and rigorous contributions to this particular line of research. I am recommending this paper for acceptance but note that this is borderline and may require reevaluation in the context of other submissions.

**Award:**

No

---

### Decision · Program_Chairs · 2022-09-14

Accept